# Interaction of 4-ethyl phenyl sulfate with bovine serum albumin: Experimental and molecular docking studies

**Payal Gulati** [1]*, **Pratima Solanki**[2], **Awadhesh Kumar Verma**[2,3], **Anil Kumar**[1]

**1** Gene Regulation Laboratory, National Institute of Immunology, New Delhi, India, **2** Special Centre for Nanoscience, Jawaharlal Nehru University, New Delhi, India, **3** School of Bioengineering and Biosciences, Lovely Professional University, Phagwara, Punjab, India

* payalgulati@nii.ac.in

**Data Availability Statement:** All relevant data are given at Mendeley data in the repository with the DOI: 10.17632/5bfvmtfxgs.1

## Abstract

4-ethyl phenyl sulfate (EPS), a protein-bound uremic toxin found in serum of patients suffering from autism spectrum disorders (ASD) and chronic kidney disease (CKD). As per recent advances in the field, gut metabolites after their formation goes to blood stream crosses blood brain barrier and causes neuro related problems. Increased levels of 4-EPS in human body causes anxiety in patients and its role remains elusive. 4-EPS interacts with serum albumin in human body and thus, a model study of interaction of BSA with 4-EPS is presented in support of it. Absorption spectroscopy result demonstrated decrease in bovine serum albumin (BSA) absorption upon interaction with increasing concentration of EPS in a range from 2 µM to 100 µM. Moreover, this interaction was confirmed by the fluorescence quenching in presence of metabolite. The change in secondary structure was demonstrated by circular dichroism, synchronous fluorescence and Fourier transform infra-red spectroscopy. Docking studies reveals binding score of −5.28 Kcal mol$^{-1}$, demarking that 4-EPS is involved in interaction with BSA via amino acid residues, forming the stable complex. This interaction study may be helpful in devising strategies for the treatment of chronic kidney disease and other neuro related diseases, by producing synthetic compound that competes with albumin binding sites to allow 4-EPS clearance from the body.

## 1. Introduction

Autism is composed of varying set of neurodevelopmental disorder attributed to early onset of lack in interaction and social communication and shows unusual behavior. It has affected 1 in 59 individuals. Certain conditions observed in ASD includes lack of communication and social interactions. Some other behavior includes restrictive interest and repetitive actions [1]. In several recent reports, one of the interesting findings is that Clostridioides species is found to be elevated in patients suffering from ASD [2–4]. Already Bolte et. al. in 1998, proposed that Clostridia has a possible role in ASD via tetanus neurotoxin (TeNT), secreted by C. tetani, which subsequently goes to central nervous system (CNS). Along with TeNT, Clostridia releases

**Funding:** MK Bhan grant with file no.- BT/HRD/MK-YRFP/50/25/2021, Department of biotechnology, New Delhi, India.

**Competing interests:** The authors have declared that no competing interests exist.

certain other toxic metabolites including phenols, para-cresol, 4-ethyl phenyl and indole derivatives [2]. Serum concentration of 4-ethyl phenyl sulfate (4-EPS) is elevated in children suffering from ASD in urine and fecal sample. Several intestinal bacteria involved in production of phenolic precursors of 4-ethylphenol are *Coriobacteriaceae*, *Enterobacteriaceae*, *Fusobacteriaceae*, and *Clostridioides* clusters I and XIVa [5]. Hsiao et al. (2013) carried out maternal immune activation (MIA) studies and found that 4-EPS serum levels and Clostridia were increased in MIA autistic mice [6]. Additionally, Hsiao et al. observed 4-EPS has structural similarity with the p-cresol and para-cresol sulfate (pCS), which indicates both have similar biosynthesis pathways.

The biosynthetic pathway of 4-EPS involves microbial degradation of aromatic amino acid (tyrosine and phenylalanine) in the gut and hepatic sulfation of the microbial metabolite 4-ethylphenol. It is produced by oxidative and reductive metabolism of tyrosine and phenylalanine through intestinal bacteria [7]. Initially, aminotransferase reaction and reductive metabolism yields 4-hydroxyphenyl propionic acid (4-HPPA) and phenylpropionic acid from tyrosine and phenylalanine, respectively. Subsequently, 4-HPPA promotes direct production of 4-ethylphenol by decarboxylation reaction [8]. These small molecules then cross intestine wall and enter blood vessels which merge with hepatic portal vein where they are directly drifted to liver. These molecules are rapidly sulphated through phenolic sulfotransferase enzymes, SULT1A1/2, that are active in organs such as brain, skin, kidney, liver and gastrointestinal tract [9]. Sulfation of these metabolites is necessary to cause their excretion through kidney as enhanced polarity makes them more water soluble. In circulation small metabolites bind with serum albumin which prevents its excretion by kidney [10, 11]. But direct association of 4-EPS is unknown yet. There is other mechanism which predicts 4-EPS retains in human body is the metabolic transformation of the ethyl chain. The ethyl chain of 4-EP is amphipathic, which makes it capable to bind with non-polar surfaces or ligands including albumin. Therefore, any modification in 4-EP is important property to reduce its binding affinity with albumin and promotes its excretion.

Serum albumin in blood takes part in several activities in human body including transport of exogenous and endogenous ligands including metal ions, fatty acid and steroids [12]. It also maintains blood pH and osmotic pressure [13]. Moreover, several phenomena such as metabolism, excretion, absorption, stability and toxicity of fatty acids are affected upon interaction with serum albumin [14]. Serum albumin shows changes in its secondary and tertiary structure, upon conjugation with small molecular weight metabolites [15]. The presence of binding sites on serum albumin with high and low affinity with which other synthetic compounds can compete for binding and facilitate the clearance of toxic metabolites from the human body.

Here, we report the binding studies of 4-EPS with BSA and their characterization with various spectroscopic methods such as synchronous fluorescence, fluorescence, UV-Visible, circular dichroism, fourier transform infra-red spectroscopy. The interaction of the conjugates with polystyrene plates were also accessed by contact angle measurement. The interaction between 4-EPS and BSA were verified by docking studies using Chem3DPro and Gaussian software and obtained binding score of -5.28 Kcal mol$^{-1}$, suggesting stable complex formation between them. Additionally, synchronous fluorescence result shows slight change in wavelength (blue shift) depicting the change in BSA configuration. There is no such literature present which illustrate about 4-EPS and BSA binding mechanism, specific binding site, conformational changes in BSA structure. This interaction study may be helpful in devising strategies for the treatment of chronic kidney disease and other neuro related diseases such as autism by producing synthetic compound that competes with albumin binding sites to allow 4-EPS clearance from the body.

## 2. Experimental section

### 2.1. Reagents and buffer

Bovine serum albumin (BSA, lyophilized), 4-ethyl phenyl sulfate (4-EPS), dialysis membrane (snake skin [TM] Dialysis Tubing, 10 K, MWtO, 16 mm), and 0.2-micron filter was procured from sigma Aldrich, mcule (Hungary), thermos Scientific, and mdi respectively. Glacial acetic acid, sodium acetate, and ethanol were procured from Merck. Acetate buffer (pH = 4.3) was prepared in Milli Q (18.2 MΩ.cm @ 25˚C), obtained from Millipore water purification system.

### 2.2. Conjugate preparation

Conjugates of BSA with 4-EPS were prepared in number of steps. In first step, the BSA solution of desired concentration of 2 µM was prepared. Subsequently, 4-EPS was dissolved in acetate buffer (pH = 4.3) in different concentration ranging from 2 µM to 100 µM, followed by the coupling reaction. Typically, coupling reaction involves the mixing of 4-EPS solution the BSA solution under controlled temperature, pH and reaction time. The complex of 4-EPS with BSA was formed in molar ratios by keeping fixed concentration of BSA (2 µM) and 4-EPS concentration was varied to 2 µM (1:1), 10 µM (1:5), 20 µM (1:10), 50 µM (1:25), 100 µM (1:50), 150 µM (1:75), 200 µM (1:100). Mixed solutions were incubated for 2 h at room temperature and for 72 h at 4˚C on the rocker shaker. To remove free (unbound) 4-ethyl phenyl sulfate from the solution, dialysis was performed. The reaction mixture was dialyzed in an acetate buffer (pH = 4.3) for consecutive three days with change of buffer after every 9 h. Aliquots of conjugates were stored at -20˚C to maintain stability.

### 2.3. Fluorescence spectroscopy

The fluorescence spectrum of BSA was measured using Cary Eclipse Fluorescence instrument, from Agilent Technologies bearing Model No.–G9800A. It is equipped with 1 cm quartz cuvette and adjustable slit width. The fluorescence of 2 µM BSA and influence of 4-EPS (2 to 200 µM in different molar ratio's) was measured in a spectral range of 300 to 600 nm wavelength at an excitation wavelength of 280 nm. The emission spectrum was measured at room temperature.

The synchronous fluorescence spectra were measured by simultaneous scan of excitation and emission monochromators. It was scanned at $\Delta\lambda$ = 15 nm and $\Delta\lambda$ = 60 nm, in a presence and absence of 4-EPS over wavelength range of 260 to 340 nm. The appropriate blank containing buffer solution spectrum was deducted from test samples to get correct fluorescence.

### 2.4. Absorption spectroscopy

The UV-Visible absorption spectrum of BSA was studied in presence and absence of gut metabolite, 4-EPS, using UV-2600, SHIMADZU (Shimadzu Corporation, in Japan) instrument. It is equipped with 1 cm quartz cuvette and parameters used for recording spectra was 600 nm/minute of scanning speed over a wavelength range of 200 to 600 nm, keeping 5 nm slit width.

### 2.5. Fourier transform infra-red spectroscopy

FTIR spectra of BSA, 4-EPS and 4-EPS:BSA conjugate were recorded over a wavelength range of 4000 to 400 cm$^{-1}$, using spectrometer unit of Perkin Elmer Spectrum 1 configuration. The scanning speed was optimized to 32 scan per second to acquire characteristic peaks. The measurement was done using liquid samples of 2 µM concentration, at room temperature.

## 2.6. Circular dichroism spectroscopy

Circular Dichroism spectra were measured on a JASCO-1500 CD spectropolarimeter instrument equipped with temperature controller (Peltier, PTC-517), water bath, and 1 mm pathlength cuvette. The measurement was done with a constant purging of $N_2$ gas inside the lamp compartment over a wavelength range of 190 to 250 nm with the fixed scanning rate of 50 nm/minute. The 0.1 cm optical pathlength of quartz cuvette was employed to measure spectra of BSA, 4-EPS and 4-EPS:BSA conjugates, with 300 μL of sample at room temperature and each spectrum was average of three successive scans. The CD spectrum of each sample was corrected by subtracting the appropriate blank corresponds to buffer. The results of CD were expressed as mean residual ellipticity (MRE), measured in deg $cm^2$ $dmol^{-1}$.

## 2.7. Contact angle measurements

Contact angle measurement of BSA, 4-EPS and 4-EPS:BSA conjugates were done to check the nature of the sample. The theta value of all the samples were recorded using drop shape analyzer [KRUSS, Germany]. The toxic behaviour of 4-EPS is a consequence of its hydrophobic nature which gets changed upon interaction with BSA molecule.

## 2.8. Theoretical modeling of ligand molecule

3-dimensional theoretical modelling of EPS molecule: a gut metabolite produced by gut microbiome, was done by using Marvin sketch software. Every time 2D and 3D refinement was performed and 3D structural confirmation of the molecule was checked by visualizing them in Marvin view. Further, the structures were optimized to transition state, and the energy was minimized sequentially by its optimization. Energy minimization experimentation was performed by utilizing Chemdraw, and Chem3DPro. The Final optimization was done using DFT approach with Gaussian 09 software with standard basis set: 6-311G and RB3LYP functions. This optimized structure was utilized for docking to monitor the interactions between EPS and BSA molecule [16].

## 2.9. Preparation of receptor molecule

3D structure of the BSA protein (RCSB PDB ID: 3V03, Source organism: Bos Taurus) with resolution 2.70 Å and 583 amino acid residues without mutation was obtained from RCSB Protein Data Bank. The protein structure was cleaned by removing all crystallographic water molecules and other crystalizing agents using AutoDockTools 4.2. Polar hydrogens including Kollmann charges were added, also Gasteiger charges were computed. The atomic type was assigned AD4 type [17].

## 2.10. Molecular docking

Finally, rigid molecular docking was performed with the optimized EPS ligand molecule using autodock 4.2 tools. Genetic algorithm, simulation program was used with a population size 300 and 100 runs. The remaining parameters were kept at their default values. The output docked file was saved as Lamarckian GA. Top 10 confirmations of protein-ligand complex were saved on based on their negative binding energy (ΔG) and RMSD values. PyMol and VMD and discovery studio software's were taken into consideration to study the interaction and binding energy of 4-EPS-BSA conjugate [18].

## 3. Results and discussion

### 3.1. Effect of 4-EPS on BSA fluorescence

Fluorescence spectroscopy is a useful technique to study the binding interaction of ligands and protein molecules. Aromatic amino acid shows useful information about folding, structure and binding interaction of proteins [19]. There are two tryptophan residues in BSA molecules, Trp 134 and Trp 213 located at hydrophilic and hydrophobic environment, respectively. BSA shows strong emission fluorescence at 348 nm, due to Trp 213, at an excitation wavelength of 280 nm. On the contrary, 4-EPS does not show any fluorescence at 348 nm upon excitation at 280 nm wavelength. Presence of fluorophore in protein molecule induces fluorescence by distinct interactions of molecules such as transfer of energy, excited state reactions, collision based quenching and complex formation with the ground state [20]. Sometimes, interaction of small molecule metabolite with the protein molecule causes changes in the fluorophore microenvironment. The fluorescent intensity of protein molecule decreases upon increasing the concentration of 4-EPS in a range from 2 µM to 100 µM (Fig 1A). The fluorescence intensity of BSA decreases with slight blue shift in wavelength around 334 nm wavelength. This suggests that there is change in the microenvironment of the trp residues by addition of 4-EPS [21].

The linearity plot of 4-EPS concentration [4-EPS] v/s fluorescence intensity [$\frac{F_o}{F}$] (Fig 1B), the stern-volmer plot, represents the 0.96 linearity within tested concentration range at room temperature. The stern-volmer equation for static quenching in terms of fluorophore, F, and quencher, Q, interaction is represented with chemical reaction having association constant, K. There are two components of Ksv: static and dynamic. In case where static quenching is dominant, association constant K is found equal between fluorophore and quencher [22–24]. The intensity of fluorescence is dependent upon final [F] and initial concentration of fluorophore, [$F_o$].$F_o$ is addition of non-fluorescent complex concentration, [FQ] and free fluorophore concentration, [F]. Stern-volmer quation is given by the relation:

$$\frac{F_o}{F} = 1 + K_{sv}\,[Q] = 1 + K_q\,\tau_o\,[Q] \tag{1}$$

The term $F$ and $F_o$ represents intensity of fluorescence in the presence and absence of the quencher respectively. $K_{sv}$ is the stern- volmer constant, [Q] is the metabolite/quencher concentration. $K_q$ (biomolecule quenching rate constant) = $K_{sv/\tau_o}$ and $\tau_o$ is a mean lifetime of BSA (fluorophore) = 10 ns [25]. The calculated $K_{sv}$ value from the plot (Fig 1C) is $1.17 \times 10^5$ Lmol$^{-1}$, and $K_q = 1.1 \times 10^{13}$ mol$^{-1}$ s$^{-1}$. In case of dynamic scattering, the maximum value of $K_q = 2.0 \times 10^{10}$ mol$^{-1}$ s$^{-1}$, it is highest collision scattering constant of small/large molecules quenchers [26]. The calculated value is greater than the reported theoretical value, suggesting intrinsic fluorescence of fluorophore (BSA) is not a consequence of dynamic quenching instead it is a result of complex formation between BSA and 4-EPS. It is a static quenching phenomenon where metabolite quenches the protein's fluorescence, thus, various binding parameters can be calculated. Fig 1C depicts linear pot between Log $\frac{F_o-F}{F}$ v/s Log [Q], with regression coefficient of 0.96 from Eq (2).

$$Log\,\frac{F_o - F}{F} = Log\,K_b + n\,Log\,[Q] \tag{2}$$

Where $K_b$ is the binding constant and n is the number of binding sites, which the slope of the plot [27].

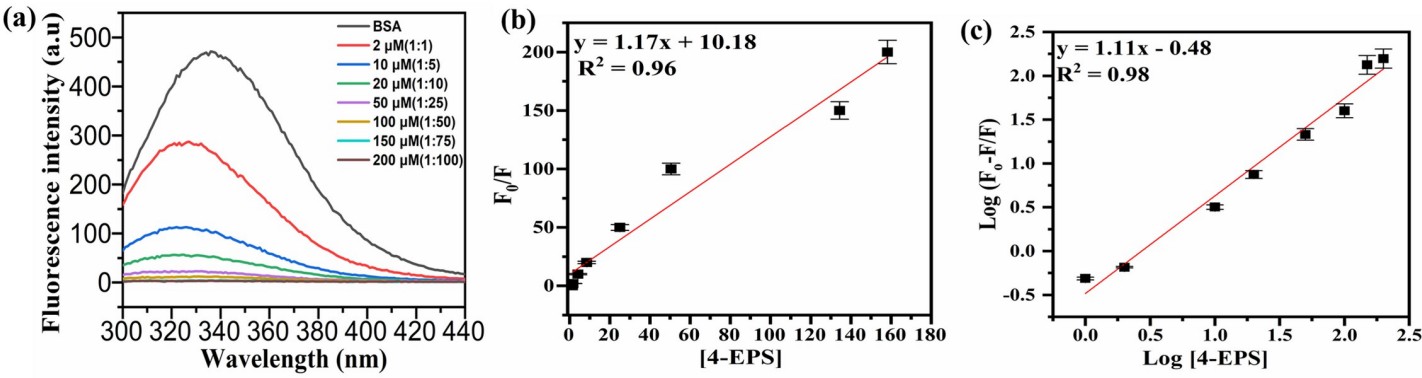

**Fig 1.** (a) Fluorescence spectrum of BSA (fixed concentration of 2 μM) in addition with varying concentration of metabolite [4-EPS] ranges from 0, 2, 10, 20, 50, 100, 150, 200 μM (A→H), respectively.; (b) Stern-Volmer plot between $\frac{F_0}{F}$ versus [4-EPS], for 4-EPS-BSA complex; and (c) Linearity plot between $Log \frac{F_0-F}{F}$ v/s log[Q] for $K_{sv}$ calculation.

Using Eq (2), the calculated values of n and $K_b$ are 1.1 and 0.48 x $10^5$ Lmol$^{-1}$, signifying, there is one strong binding site present on BSA for metabolite. Thus, resulting in strong binding interaction between 4-EPS and BSA.

Thermodynamical parameters were calculated using relations:

$$\Delta G^{\circ} = -RT \ln K_b \tag{3}$$

$$\Delta H^{\circ} = \frac{\partial \left( \Delta G^{\circ}/T \right)}{\partial (1/T)} \tag{4}$$

$$\Delta S^{\circ} = \frac{\left( \Delta H^{\circ}/\Delta G^{\circ} \right)}{T} \tag{5}$$

The nature of interactions including hydrophobic, hydrogen bond, van der Waals or and electrostatic interaction between BSA and 4-EPS can be predicted from the signs and magnitudes of the thermodynamical parameters. Ross and Subramanian have analysed which type of interaction takes place with proteins with respect to the signs and magnitudes of the thermodynamical parameters ($\Delta H^{\circ}$ and $\Delta S^{\circ}$) [28]. Different conditions involved are: (a) when $\Delta H^{\circ} > 0$ and $\Delta S^{\circ} > 0$, then the dominating force is hydrophobic; (b) when $\Delta H^{\circ} < 0$ and $\Delta S^{\circ} < 0$, then the dominating force is Vander Waals and hydrogen bonding; (c) when $\Delta H^{\circ} < 0$ and $\Delta S^{\circ} > 0$, then electrostatic force will dominate. The values of these parameters $\Delta G^{\circ}, \Delta H^{\circ}$, and $\Delta S^{\circ}$ were calculated using Eqs (3), (4), and (5), respectively and obtained as -26.257 KJmol$^{-1}$, 28.74 KJmol$^{-1}$, and 8.49 Jmol$^{-1}$K$^{-1}$, respectively (given in Table 1). The negative sign of $\Delta G$ indicates the binding of BSA and 4-EPS is a spontaneous process. Binding of BSA and 4-EPS is an endothermic process which is represented by positive value of entropy. The attained values of $\Delta H^{\circ}$ and $\Delta S^{\circ}$ were found to be greater than zero which means main dominating force involved is hydrophobic [29]. These results suggest that secondary structure of BSA undergoes changes due to 4-EPS. These results were confirmed by CD measurements.

**Table 1. Thermodynamical parameters of 4-EPS and BSA interaction.**

| Binding Entities | $\Delta G^{\circ}$ | $\Delta H^{\circ}$ | $\Delta S^{\circ}$ | Molecular Interaction |
|---|---|---|---|---|
| 4-EPS-BSA | -26.257 KJmol$^{-1}$ | 28.74 KJmol$^{-1}$ | 8.49 Jmol$^{-1}$K$^{-1}$ | $\Delta H^{\circ}$ and $\Delta S^{\circ}$ >0: suggest hydrophobic force. |

## 3.2. UV-Visible spectroscopic results

The absorption of BSA (black), 4-EPS (red), and 4-EPS-BSA conjugate (blue), is shown in uv-vis spectrum (Fig 2A). This figure depicts the significant absorption intensity of BSA in comparison to 4-EPS at 2 μM concentration.

Absorption intensity of BSA decreases at 280 nm upon addition of 4-EPS in increasing concentration from 2 μM to 200 μM [30], from Fig 2B. This supports the fluorescence quenching attained in PL spectrum of 4-EPS- BSA binding interaction (Fig 1B). The linearity plot between $A/(A_o—A)$ v/s 1/ [4-EPS], shows linearity of 0.99 and obtained detection limit of 0.57 μM. BSA absorption peak at 280 nm shows the presence of aromatic amino acids including Tyr, Typ and Phe residues in it and the intensity of absorption shows the counts of aromatic amino acids [31]. Therefore, this absorption spectroscopy is a useful technique for protein quantification on the basis of absorption intensity. The absorption intensity at 280 nm of aromatic amino acid is a result of side chain ring structure in their R group and π electrons delocalization. The Benesi-Hildebrand plot between $A/(A_o—A)$ v/s 1/ [4-EPS], shown in Fig 2C, where slope of the graph gives the values of number of binding sites present on the proteinaceous species for the metabolite molecule [32]. According to the plot, there are 2 binding sites located on the serum albumin, which is in good agreement with the computational data.

## 3.3. Fourier transform infra-red spectroscopy studies

Fourier transform infra-red spectroscopy is utilized for identification of the functional bond through characteristics peaks attained in the spectrum corresponding to the wavenumber. Complex formed between BSA and 4-EPS was confirmed on comparing the spectrum of pure 4-EPS and BSA with their conjugate in acetate buffer with pH = 4.3. BSA spectrum represents two characteristic peaks of the molecules at 1543 $cm^{-1}$ and 1641 $cm^{-1}$ corresponding to flexural vibration adsorption of Amide II (−NH−) and Amide I (−NH$_2$), respectively [33], shown in Fig 3A. In 4-EPS spectrum (in Fig 3B), the peaks obtained at 1461 $cm^{-1}$ and 2852 $cm^{-1}$, represents the anti-symmetric deformation in -CH$_3$; and -CH anti-symmetric and symmetric stretch in aliphatic compounds, respectively is may be due to the presence of ethyl group in the compound. The benzene ring stretch at 1508 $cm^{-1}$ is may be due to fermi resonance with overtone and combinations [34]. The peak position at 3610 $cm^{-1}$ corresponds to the stretch of -OH group located at the para position of the benzene ring and at 1375 $cm^{-1}$ corresponds to the

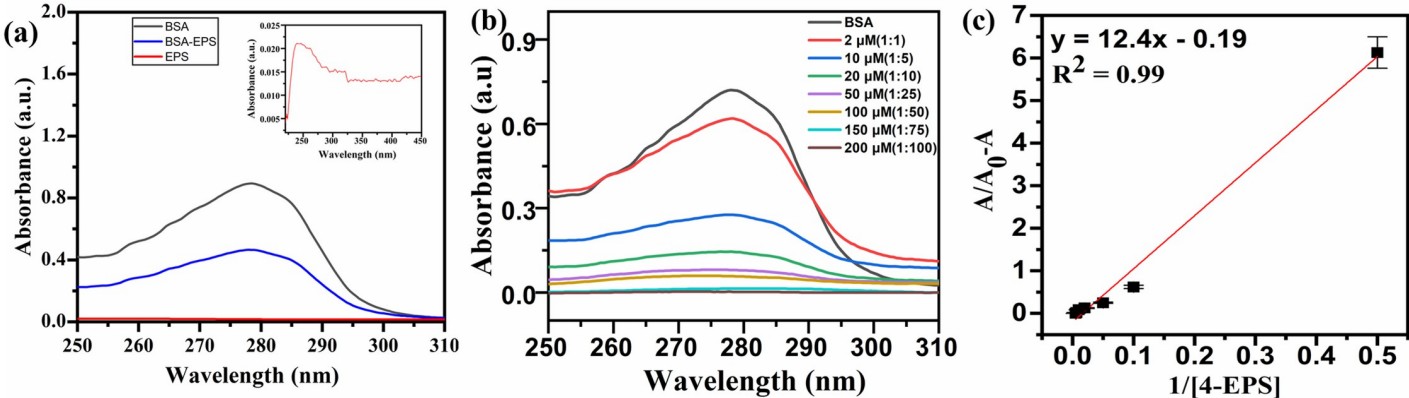

**Fig 2.** (a) UV-Visible absorption spectrum representing BSA, 4-EPS and 4-EPS-BSA conjugate at concentration of 2 μM in acetate buffer (pH = 4.3); (b) Absorption spectrum representing 2 μM BSA with different concentration of metabolite from 0, 2, 10, 20, 50, 100, 150 and 200 μM ($A{\rightarrow}H$); (c) Benesi-Hildebrand plot between $A/(A_o$-A) v/s 1/ [4-EPS].

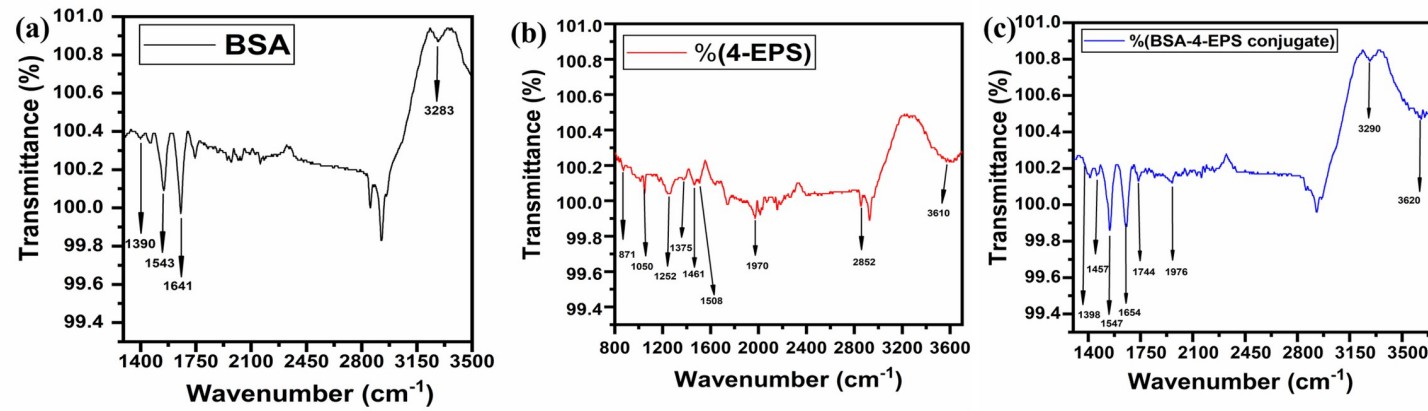

**Fig 3.** Fourier transform infra-red spectrum of: (a) BSA at 2 μM concentration in acetate buffer pH = 4.3; (b) 4-EPS at 2 μM concentration in acetate buffer pH = 4.3; (c) 4-EPS-BSA conjugate.

anti- symmetric stretch in $SO_2$, which suggest the sulfation of 4-ethyl phenyl group and $C = O$ stretch in esters at 1744 cm$^{-1}$ is due to the solvent of 4-EPS solution. 4-EPS and BSA complex was confirmed by formation of covalent bond with $C = O$ group of 4-EPS and $N—H$ group of BSA, represented by the deformations in the amide bond, at 1547 cm$^{-1}$ and 1654 cm$^{-1}$ are characteristic peaks (shown in Fig 3C), as represented in Table 2.

## 3.4. Conformation studies

### 3.4.1. Circular dichroism studies.
It is sensitive tool which demonstrate any deviation from the secondary and tertiary structure of the protein molecules upon interaction with the

**Table 2. Fourier transform infra-red peaks at different wavenumber and their assignment.**

| Entities | Wavenumber | Assigned To |
|---|---|---|
| **EPS** | 871 | Alkenes |
| | 1050 | -$OH$ bond |
| | 1252 | Skeletal vibration of tert-butyl in hydrocarbons |
| | 1375 | Anti- symmetric stretch in $SO_2$ |
| | 1461 | Anti-symmetric deformation in -$CH_3$ in aliphatic compounds |
| | 1508 | Benzene ring stretch in aromatic compounds |
| | 1970 | $C = C = C$ anti-symmetric stretch in alkenes |
| | 2852 | CH anti-symmetric and symmetric stretch in aliphatic compounds |
| | 3610 | OH stretch in alcohols and phenols |
| **EPS-BSA Conjugate** | 1398 | $COO^-$ symmetric stretch in carboxylic acid salts |
| | 1457 | Anti-symmetric deformation in -$CH_3$ in aliphatic compounds |
| | 1547 | $NH$ deformation in secondary amides, amide-II band. |
| | 1654 | $NH$ deformation in primary amides |
| | 1744 | $C = O$ stretch in esters |
| | 1976 | $C = C = C$ anti-symmetric stretch in alkenes |
| | 3290 | $NH$ stretch in secondary amides |
| | 3620 | OH stretch in alcohols and phenols |
| **BSA** | 1390 | $CH_3$ and $CH_2$ stretch |
| | 1543 | Amide II |
| | 1641 | Amide I |
| | 3283 | Amide A |

ligand molecules [35]. A BSA is a bulky proteinaceous molecule representing characteristic peaks at θ = 210 nm and 222 nm in a uv region corresponds to the alpha-helical structure of protein and negative sign of the peaks shows the $n \rightarrow \pi^*$ *transition* for the peptide bond of *α helix*. The CD spectrum of the BSA interaction with 4-EPS, depicts decrease in band intensity from A→H as the metabolite concentration was increased, represented in Fig 4. Upon addition of metabolite in increasing concentration, keeping fixed concentration of BSA, did not show any shift in the peaks which clearly suggests dominant *α-helix* structure of BSA is maintained.

**3.4.2. Synchronous fluorescence spectroscopic analysis.** Synchronous fluorescence spectroscopy is a simplest technique to identify any change in the core molecular micro-environment and to measure fluorescence quenching [36]. This tool helps to analyse the conformational changes in proteins upon interaction with other molecules (ligands). It can provide information about the molecules micro-environment near the chromophore molecules. This method discusses about the amino acid residues environment by measuring the possible shift in the maximum wavelength of synchronous fluorescence $\lambda_{SFS.max}$. The shift in maximum wavelength corresponds to polarity change around the chromophore molecule where change at Δλ = 15 nm corresponds to tyrosine residues and at Δλ = 60 nm, then change is around tryptophan residues. The change in the wavelength from 287.9 nm to 282.37 nm, corresponds to the blue shift in maximum emission λ with increasing metabolite concentration from 2 μM to 200 μ M, (shown in Fig 5A) thus there is conformation change in secondary structure of BSA and suggest more hydrophobic environment of tyrosine residues [37]. While in Fig 5B, the maximum emission wavelength showed slight change in wavelength from 287.8 nm to 284.2 nm and this revealed there is slight change in tryptophan residues. Additionally, the decrease in fluorescence intensity was found at both the wavelengths (Δλ = 15 nm, and Δλ = 60 nm), suggest fluorescence quenching in the interaction of 4-EPS with BSA.

## 3.5. Contact angle studies

Contact angle studies of BSA, 4-EPS and 4-EPS-BSA conjugate is shown in Fig 6A and with increasing metabolite concentration is shown in Fig 6B. The study was carried out on the ELISA plate which is polystyrene in nature. The hydrophobic nature of the plate is suitable for

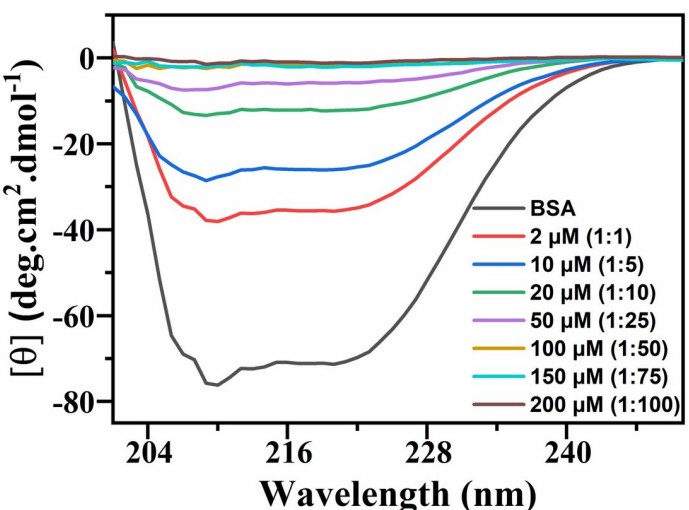

**Fig 4.** Circular dichroism spectrum of 4-EPS-BSA conjugate, where BSA is at a fixed concentration of 2 μ*M* and 4-EPS concentration ranges from 0, 2, 10, 20, 50, 100, 150, 200 μM (*A→H*), respectively, at room temperature.

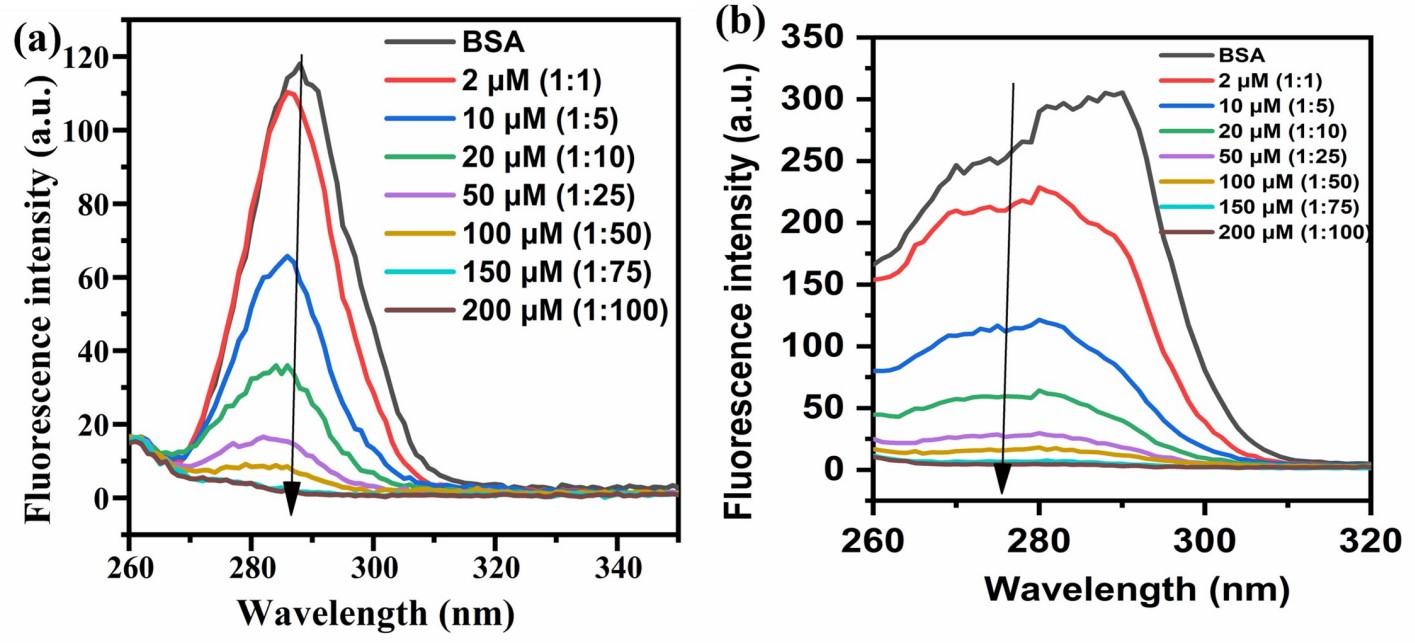

**Fig 5.** Synchronous fluorescence spectrum of 4-EPS-BSA: (a) at $\Delta\lambda = 15$ nm, and (b) at $\Delta\lambda = 60$ nm.

binding with proteins and ligands molecules. Therefore, carboxylic and amine groups can easily form bond with the plate as represented by the θ values.

4-EPS is an amphiphile molecules that have both hydrophobic and hydrophilic domains [1]. The hydrophobic domain of this molecule allows it to be solubilized in water by forming micelle and bilayers. Moreover, the folded stable structure of albumin protein is due to the hydrophobic core where hydrophobic side chains from water are embedded inside hydrophobic core. The obtained θ values from contact angle studies clearly depicts that on increasing the metabolite concentration the hydrophobicity of the complex increases (shown in part Fig 6B (a to g)), thus representing a stable conjugate formation. The increase in hydrophobicity is represented by the increasing θ value from 1:1 conjugate to 1:100, mentioned in Fig 6B (a) to (g).

### 3.6. Docking results

**3.6.1. In silico approach for modelling and docking studies on 4-EPS-BSA complex.** The ground state configuration 4-EPS as ball and stick model, is shown in Fig 7A. The grey color ball represents the C atoms, white color ball as hydrogen atoms, red color as oxygen atoms and dark yellow color as sulphur atom in configuration of 4-EPS. Entire structure is having one hexagonal benzene ring attached with oxygen atom (represented through red color) which is further attached to Sulphur atoms. Ground state configuration of 4-EPS shown in Fig 7A before optimization. Fig 7B shows the structure of 4-EPS, generated from the structure drawn after optimization and energy minimization using Gaussian09 software through DFT approach.

The detail information for bond length, bond angle and dihedral angle for each atom of 4-EPS molecule including energy minimization and RMS gradient normalization with optimization step number has been shared in supplementary data in S1, S2 Tables and S1 Fig in S1 File, respectively. After optimization, the bond lengths in 4-EPS molecular structure shows no change while bond angle shows slight modifications.

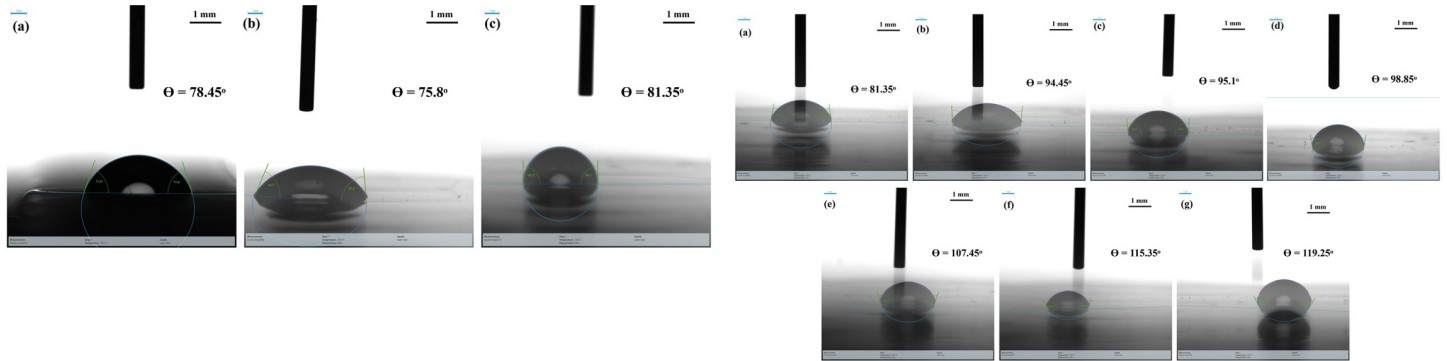

**Fig 6.** A. Contact angle studies: (a) pure BSA at 2µM concentration in acetate buffer; (b) pure 4-EPS at 2 µM concentration in acetate buffer. B. Contact angle studies of 4-EPS-BSA conjugates with fixed 2 µM concentration of BSA and different concentration of 4-EPS ranges from 2, 10, 20, 50, 100, 150, and 200 µM (a→g).

**3.6.2. Docking of 4-EPS with bovine serum albumin.** In the current work we have also utilized computational approaches like molecular docking which could provide the molecular insights into the interaction between 4-EPS and bovine serum albumin which may not be accessible from experiment alone. Table 3 showing the rank wise 10 different confirmations of 4-EPS-BSA complex. Rank one showing the best binding energy -5.28 Kcal/mol with cluster RMSD 0.00 and reference RMSD103.49 of 4-EPS-BSA complex.

The docked structure of 4-EPS with bovine serum albumin, shown in Fig 8, was drawn using pymol software in which some part of the amino acid residue is showing involvement of non-covalent interaction with 4-EPS and it clearly describes that 4-EPS is bound to the protein surface in its active site. THR 419, LYS 499 and LYS 533 and ARG 409 was involved in conventional hydrogen bonding, while TYR 496 involved in Alkyl interaction. LYS 419 also involved in pie alkyl interaction.

From the above docking study, it is clear that the EPS is having good binding affinity with bovine serum albumin and 4-EPS is binding to the surface of bovine serum albumin in 4-EPS-BSA complex.

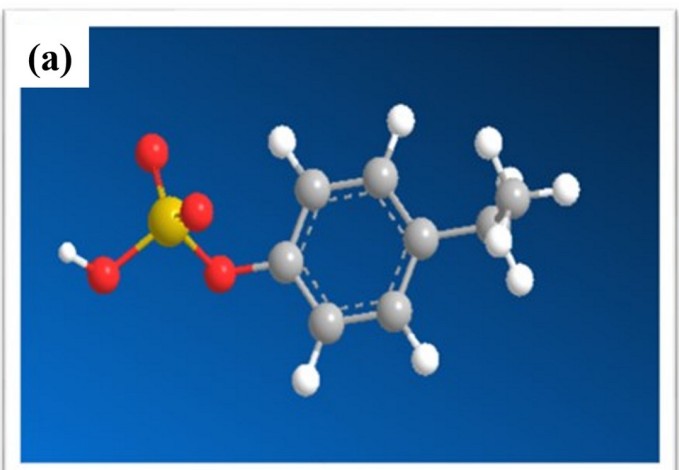
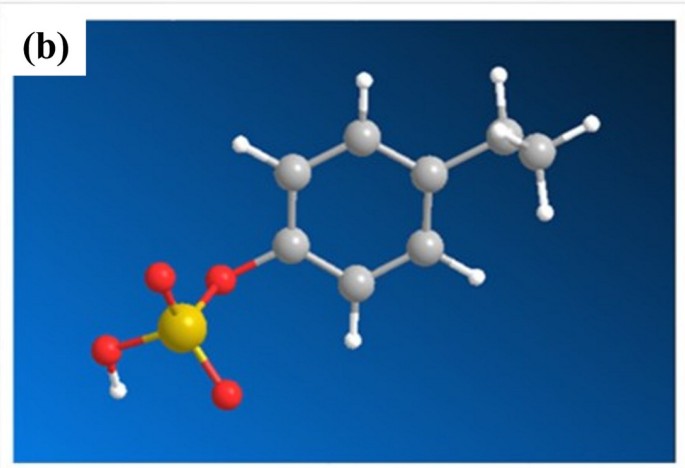

**Fig 7.** (a) 3D ground state configuration 4-EPS before optimization (b) after optimization.

**Table 3. Showing the rank wise confirmation of 4-EPS-BSA complex with corresponding energy.**

| Rank | Sub-Rank | Run | Binding Energy | Cluster RMSD | Reference RMSD | Grep Pattern |
|---|---|---|---|---|---|---|
| 1 | 1 | 1 | -5.28 | 0.00 | 103.49 | RANKING |
| 2 | 1 | 3 | -4.97 | 0.00 | 70.11 | RANKING |
| 2 | 2 | 5 | -4.94 | 0.76 | 69.94 | RANKING |
| 2 | 3 | 8 | -4.89 | 0.88 | 69.90 | RANKING |
| 2 | 4 | 10 | -4.78 | 0.79 | 69.95 | RANKING |
| 3 | 1 | 6 | -4.96 | 0.00 | 84.80 | RANKING |
| 4 | 1 | 4 | -4.73 | 0.00 | 105.41 | RANKING |
| 4 | 2 | 2 | -4.42 | 1.16 | 105.24 | RANKING |
| 5 | 1 | 7 | -4.01 | 0.00 | 103.39 | RANKING |
| 6 | 1 | 9 | -3.19 | 0.00 | 113.83 | RANKING |

## 4. Conclusion

A brief of interaction studies of 4-EPS with BSA have shown with the help of spectroscopic tools and computational methods. Results obtain from both the techniques suggest good binding affinity between them. Absorption spectroscopy reveals the decrease in absorption intensity of BSA upon interaction with 4-EPS which was confirmed by fluorescence quenching in the emission spectrum of BSA upon addition of increasing concentration of 4-EPS. The calculated LOD from absorption linearity graph is 0.57 μM in a concentration range from 2 μM to 200 μM of 4-EPS. The conformation change was investigated with the help of circular dichroism and synchronous fluorescence spectroscopy. CD spectroscopic results suggest that the *α-helix* structure is maintained but synchronous fluorescence studies suggest the change microenvironment of the tyrosine and tryptophan residues of BSA molecules. The complex is stabilized with both hydrophobic interaction and hydrogen binding. Uremic toxin, 4-EPS is believed to bind with serum albumin through ethyl phenyl chain and if this can be modified by changing its polarity can facilitate its removal from the body. In this regard, this study

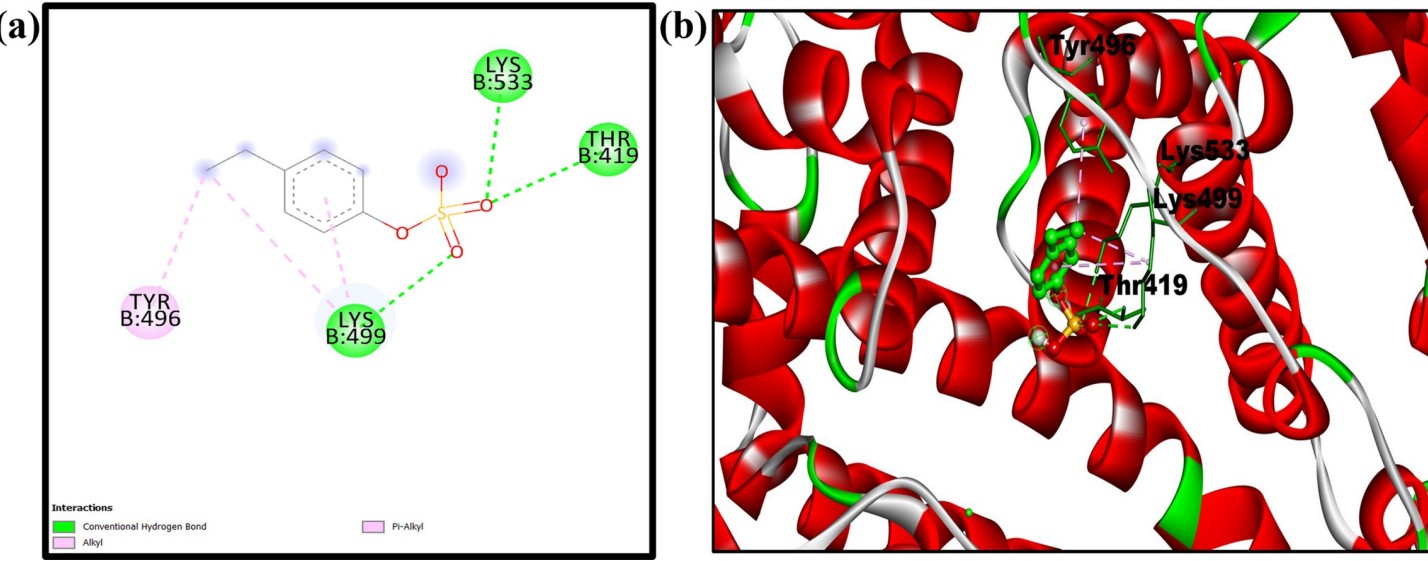

**Fig 8.** Shows 3D view of involvement of non-covalent interaction of 4-EPS-BSA complex; and (b) shows the 2D interactions of 4-EPS with BSA, predicted by the docking study.

provides an insight to check the binding affinity between 4-EPS and BSA and after modifying 4-EP chain to check reduced affinity between them. This technique will help in treatment of autism in patients.

## Supporting information

**S1 File. Insilco analysis of interaction between BSA and 4-EPS: S1 Fig relates to energy minimization and RMS gradient normalization for 4-EPS with optimization stem number.** S1 Table relates to ground state configurations of 4-EPS with every atom's spatial arrangement: bond length and angle. S2 Table relates to dipole moment, optimized energy Value and RMS gradient normalization value for optimized EPS molecule.
(DOCX)

## Acknowledgments

I acknowledge the administrative and infrastructure support from NII for providing me the platform to accomplish the research work. I also acknowledge central instrumentation facility of special centre for nanoscience, JNU for instrumentation support.

## Author Contributions

**Conceptualization:** Payal Gulati, Pratima Solanki, Awadhesh Kumar Verma.

**Data curation:** Pratima Solanki, Awadhesh Kumar Verma.

**Formal analysis:** Awadhesh Kumar Verma, Anil Kumar.

**Funding acquisition:** Payal Gulati.

**Investigation:** Payal Gulati.

**Methodology:** Payal Gulati.

**Project administration:** Payal Gulati.

**Resources:** Pratima Solanki.

**Supervision:** Payal Gulati, Anil Kumar.

**Writing – original draft:** Payal Gulati, Awadhesh Kumar Verma.

**Writing – review & editing:** Pratima Solanki, Anil Kumar.

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
