## [Decision Letter · Decision Letter 0]

11 Jun 2024

PONE-D-24-20499Interaction of 4-ethyl phenyl sulfate with BSA: Experimental and molecular docking studiesPLOS ONE

Dear Dr. Gulati,

Thank you for submitting your manuscript to PLOS ONE. After careful consideration, we feel that it has merit but does not fully meet PLOS ONE’s publication criteria as it currently stands. Therefore, we invite you to submit a revised version of the manuscript that addresses the points raised during the review process.

We look forward to receiving your revised manuscript.

Kind regards,

Sabato D'Auria

Academic Editor

PLOS ONE

Journal Requirements:

2. Please note that PLOS ONE has specific guidelines on code sharing for submissions in which author-generated code underpins the findings in the manuscript. In these cases, we expect all author-generated code to be made available without restrictions upon publication of the work. 

Please review our guidelines at https://journals.plos.org/plosone/s/materials-and-software-sharing#loc-sharing-code and ensure that your code is shared in a way that follows best practice and facilitates reproducibility and reuse.

"MK Bhan grant with file no.- BT/HRD/MK-YRFP/50/25/2021, Department of biotechnology, New Delhi, India."

"We greatly acknowledge the funding support from DBT, the funding agency. This proposed project work was funded by MK Bhan grant with file no.- BT/HRD/MK-YRFP/50/25/2021, Department of biotechnology, New Delhi, India."

Please note that funding information should not appear in the Acknowledgments section or other areas of your manuscript. We will only publish funding information present in the Funding Statement section of the online submission form. Please remove any funding-related text from the manuscript. 

6. Thank you for stating the following in your Competing Interests section: "None"

7. Please provide a complete Data Availability Statement in the submission form, ensuring you include all necessary access information or a reason for why you are unable to make your data freely accessible. If your research concerns only data provided within your submission, please write "All data are in the manuscript and/or supporting information files" as your Data Availability Statement.

8. When completing the data availability statement of the submission form, you indicated that you will make your data available on acceptance. We strongly recommend all authors decide on a data sharing plan before acceptance, as the process can be lengthy and hold up publication timelines. Please note that, though access restrictions are acceptable now, your entire data will need to be made freely accessible if your manuscript is accepted for publication. This policy applies to all data except where public deposition would breach compliance with the protocol approved by your research ethics board. If you are unable to adhere to our open data policy, please kindly revise your statement to explain your reasoning and we will seek the editor's input on an exemption. Please be assured that, once you have provided your new statement, the assessment of your exemption will not hold up the peer review process.

Reviewers' comments:

Reviewer's Responses to Questions

**Comments to the Author**

1. Is the manuscript technically sound, and do the data support the conclusions?

Reviewer #1: Partly

Reviewer #2: Partly

2. Has the statistical analysis been performed appropriately and rigorously? 

Reviewer #1: Yes

Reviewer #2: Yes

3. Have the authors made all data underlying the findings in their manuscript fully available?

Reviewer #1: Yes

Reviewer #2: Yes

4. Is the manuscript presented in an intelligible fashion and written in standard English?

Reviewer #1: Yes

Reviewer #2: No

5. Review Comments to the Author

Reviewer #1: The manuscript by Payal Gulati et al. entitled “Interaction of 4-ethyl phenyl sulfate with BSA: Experimental and molecular docking studies” describes the interaction of BSA with 4-EPS using different techniques.

The authors conclude that this interaction study may be helpful in devising strategies for the treatment of chronic kidney disease and other neuro related diseases wherein interaction of 4-EPS and BSA matters.

In my opinion the work should be improved. Data presented demonstrate the link between BSA and 4-EPS but it was already known news. There are no predictions of how this link might be severed. The work does not bring anything new to readers.

I admit this manuscript with major revision.

In particular:

- I would like to suggest improving the introduction. The introduction does not seem to have a linearity, the different parts do not seem connected to each other.

- The authors declare that the BSA-4-EPS conjugate was purified from unreacted material and by-products, but they don't describe how they do it (2.2 Conjugate preparation paragraph).

- I would suggest including the control of the experiments shown in Figure 1 and Figure 2. I would like to suggest demonstrating that the decrease in fluorescence intensity is due to the interaction with the molecule and not to a change in volume or other.

- In figures 1a and 2b I would like to suggest inserting the legend with the concentrations for a more immediate view.

- In figure 3, EPS is shown and not 4-EPS and is not readable. To compare the graphs, they should all start from the same percentage of transmittance.

- The resolution of the graphics is very bad

- Why were the experiments carried out at pH 4.3?

- In figure 7, the letters a and b cannot be read.

- The authors in the abstract declare that: increased level of 4-EPS in human body has also been found implicated in anxiety; in the conclusion they declare that: This technique will help in treatment of autism in patients. There is no linearity.

Reviewer #2: In the manuscript “Interaction of 4-ethyl phenyl sulfate with BSA: Experimental and molecular docking studies” the authors study the interaction of the e4-ethyl phenyl sulfate with the bovine serum albumin.

In my opinion, this work should be of interest to the readers of the Journal but I recommend the publication of this work after a major revision. In particular:

1) The title of the article contains the abbreviation BSA and not its full name. Please, the authors modify it.

2) In the abstract, is not clear the main aim of the article.

3) The introduction section is too long and it is difficult to easily understand the aim of the work. The authors declare that the selected compound is associated with autism disorder so, why the authors study this interaction with BSA and not with HSA? Please, the authors clarify it.

4) The sentence in paragraph 2.2 “confirmation of the successful…studies” should be removed because is not needed in the MM section.

5) In the MM the authors include paragraph 2.2 defined “conjugate preparation” in which they described the sample preparation (BSA + 4-ethyl phenyl sulfate) before the spectroscopy characterization. The authors describe the incubation of the BSA with different analyte concentrations. The protocol reported is not clear. Please, the authors clarify it.

Why the authors perform the incubation with an increased concentration of the analytes at pH 4.3? Have the authors studied the effect of this pH on the BSA structure? What is the effect on the binding? Have the authors information on the binding at neutral pH instead of low pH? Please, the authors clarify it.

6) In the fluorescence spectroscopy paragraph reported in the MM section, the authors have not reported the excitation wavelength value used in the steady-state experiments. Please, the authors provide it.

7) Have the authors verified the optical density (OD) of the BSA at the concentration of 2uM? This value should be between 0.05-0.1 OD at the excitation value to avoid the inner filter effect. Have the authors control it?

8) What are the positive and negative controls in the experiments? Please, the authors provide it.

9) Paragraph 3.3 starts with the figure legend, table 1, and then describes the FT-IR results. Please, the authors re-organize this paragraph. Similar problems there are for the paragraphs 3.4.1, 3.4.2, and 3.5. Please, the authors fix it.

10)In the material methods section the authors describe the absorption spectroscopy experiments and then the fluorescence spectroscopy experiments. Why in the “results and discussion” section the authors show before the florescence results and then the absorption results? This order does not have sense. Please, the authors modify it.

10) In the “results and discussion” section the authors discuss the fluorescence experiments and report the excitation value used (280 nm). At this wavelength are excite both the tryptophan and tyrosine residues present in BSA and not only the tryptophan as discussed by the authors. Normally, the binding interaction of ligands and protein is followed as a variation of the tryptophan microenvironment by excitation at 295 nm. So, the results obtained in this condition (excitation at 280nm) need to be discussed considering both fluorescence contributions (tryptophan and tyrosine). Please the authors repeat the experiments by excite the BSA at 295 nm and compare the obtained results with the data reported in the manuscript.

11) Why the fluoresce emission was recorded in the range of 300-600 nm? The second pick present in the spectra, that appears at 560 nm (double of 280nm), is the harmonic pick of the excitation (see Figure 1A). This has no sense. Please the author modifies Figure 1.

12) In Figures 2 A and 2B the wavelength range is not the same. Please, the authors fix it.

12)From the quenching experiments the authors have calculated the thermodynamic parameters of the binding between BSA and the 4-ethyl phenyl sulfate. Please, to allow the readers to better understand these results include the obtained results in a table and discuss it with the type of molecular interaction identified.

13) All the mathematical equations (Stern-Volmer, thermodynamic equations, etc.) used and reported in the “results and discussion” section should be removed and included only in the “materials and method” section. Please the authors provide it.

14) All the figure's legends should be increased. Please, the authors provide it.

15) Table 2 appears in a different layout respect to table 1. Please, the authors re-organize it.

6. PLOS authors have the option to publish the peer review history of their article (what does this mean?). If published, this will include your full peer review and any attached files.

Reviewer #1: No

Reviewer #2: No

---

## [Author Response · Author response to Decision Letter 0]

1 Aug 2024

Response to Reviewers

Date: 26/07/2024

To,

The Editor,

PLOS ONE

Subject: Submission of revised manuscript “Ref. No.: PONE-D-24-20499. Article title: Interaction of 4-ethyl phenyl sulfate with bovine serum albumin: Experimental and molecular docking studies”.

Dear Editor,

Thank you for your useful comments and suggestions regarding the scientific and technical aspects of our manuscript. We have modified the manuscript accordingly, and the detailed corrections are listed below point by point:

Reply to the academic editor and reviewer(s) comments

Journal Requirements:

Ans: As Advised, the manuscript is revised according to the PLOS ONE template.

2. Please note that PLOS ONE has specific guidelines on code sharing for submissions in which author-generated code underpins the findings in the manuscript. In these cases, we expect all author-generated code to be made available without restrictions upon publication of the work. 

Please review our guidelines at https://journals.plos.org/plosone/s/materials-and-software-sharing#loc-sharing-code and ensure that your code is shared in a way that follows best practice and facilitates reproducibility and reuse.

Ans: Thank you for the query. But this is for coding-based manuscript and this work includes the interaction study.

Ans: As advised, this will be taken care while uploading the manuscript.

"MK Bhan grant with file no.- BT/HRD/MK-YRFP/50/25/2021, Department of biotechnology, New Delhi, India."

Ans: As advised, we will update in cover letter that funders had provided fund for this study, but funders had no role in study design, data collection and analysis, decision to publish or preparation of manuscript. We greatly acknowledge DBT for providing funds to conduct this study.

"We greatly acknowledge the funding support from DBT, the funding agency. This proposed project work was funded by MK Bhan grant with file no.- BT/HRD/MK-YRFP/50/25/2021, Department of biotechnology, New Delhi, India."

Please note that funding information should not appear in the Acknowledgments section or other areas of your manuscript. We will only publish funding information present in the Funding Statement section of the online submission form. Please remove any funding-related text from the manuscript. 

Ans: As advised, the change is made in the revised manuscript. The financial disclosure is separated from acknowledgement.

6. Thank you for stating the following in your Competing Interests section: "None"

Ans: As advised, the competing interest statement is included in the cover letter.

7. Please provide a complete Data Availability Statement in the submission form, ensuring you include all necessary access information or a reason for why you are unable to make your data freely accessible. If your research concerns only data provided within your submission, please write "All data are in the manuscript and/or supporting information files" as your Data Availability Statement.

Ans: As advised, all the authors agreed to make data available as per the journal’s policy information and will be ensured in the submission form. 

8. When completing the data availability statement of the submission form, you indicated that you will make your data available on acceptance. We strongly recommend all authors decide on a data sharing plan before acceptance, as the process can be lengthy and hold up publication timelines. Please note that, though access restrictions are acceptable now, your entire data will need to be made freely accessible if your manuscript is accepted for publication. This policy applies to all data except where public deposition would breach compliance with the protocol approved by your research ethics board. If you are unable to adhere to our open data policy, please kindly revise your statement to explain your reasoning and we will seek the editor's input on an exemption. Please be assured that, once you have provided your new statement, the assessment of your exemption will not hold up the peer review process.

Ans: As advised, Authors agree to the journal’s data availability policy. All authors are agreed for journal data availability policy.

Ans: As advised, captions of supporting information file is included in the revised manuscript.

Reviewer #1: The manuscript by Payal Gulati et al. entitled “Interaction of 4-ethyl phenyl sulfate with BSA: Experimental and molecular docking studies” describes the interaction of BSA with 4-EPS using different techniques.

The authors conclude that this interaction study may be helpful in devising strategies for the treatment of chronic kidney disease and other neuro related diseases wherein interaction of 4-EPS and BSA matters.

In my opinion the work should be improved. Data presented demonstrate the link between BSA and 4-EPS but it was already known news. There are no predictions of how this link might be severed. The work does not bring anything new to readers.

I admit this manuscript with major revision.

In particular:

- I would like to suggest improving the introduction. The introduction does not seem to have a linearity, the different parts do not seem connected to each other.

Ans: As advised, the introduction has been modified to be crisper, and connections between paragraphs have been established. The modified introduction is incorporated in the revised manuscript.

- The authors declare that the BSA-4-EPS conjugate was purified from unreacted material and by-products, but they don't describe how they do it (2.2 Conjugate preparation paragraph).

Ans: As advised, the sentence is modified for representing the purpose of dialysis and is given below for your ready reference:

To remove free (unbound) 4-ethyl phenyl sulfate from the solution, dialysis was performed. The reaction mixture was dialyzed in an acetate buffer (pH=4.3) for consecutive three days with change of buffer after every 9 h.

- I would suggest including the control of the experiments shown in Figure 1 and Figure 2. I would like to suggest demonstrating that the decrease in fluorescence intensity is due to the interaction with the molecule and not to a change in volume or other.

Ans: Thank you for the query, I would like to refine the language to bring clarity. As mentioned in the section 2.2 conjugate formation, the complex of 4-EPS with BSA was formed in molar ratios by keeping fixed concentration of BSA (2 μM) and 4-EPS concentration was varied to 2 μM (1:1), 10 μM (1:5), 20 μM (1:10), 50 μM (1:25), 100 μM (1:50), 150 μM (1:75), 200 μM (1:100). This statement is incorporated in the revised manuscript.

The experiments (UV-Vis and fluorescence study) were also performed in 1:1 ratio with fixed concentration of BSA and varied concentration of 4-EPS (2 μM to 200 μM) but linearity of the graph was compromised therefore, the work was again done in different ratios, as mentioned above. The graph of this study is given below for your ready reference:

Figure 1: Results of 4-EPS and BSA complex in 1:1 - (a) Fluorescence spectrum of BSA (fixed concentration of 2 μM) in addition with varying concentration of metabolite [4-EPS] ranges from 0, 2, 10, 20, 50, 100, 150, 200 μM (A→H), respectively; (b) Absorption spectrum representing 2 μM BSA with different concentration of metabolite from 0, 2, 10, 20, 50, 100, 150 and 200 μM (A→H).

- In figures 1a and 2b, I would like to suggest inserting the legend with the concentrations for a more immediate view.

Ans: As advised, the legends are inserted with the concentration for more immediate view and incorporated as revised figure and is given below for your ready reference:

- In figure 3, EPS is shown and not 4-EPS and is not readable. To compare the graphs, they should all start from the same percentage of transmittance.

Ans: As advised changes are made in the graph and incorporated as revised figure 3 and is given below for your ready reference:

- The resolution of the graphics is very bad

Ans: As advised, all the figures are again saved as TIF format.

- Why were the experiments carried out at pH 4.3?

Ans: According to one of the references, Milad Moradi et. al. suggest that PCS fluorescence intensity is enhanced in acidic pH [1].

- In figure 7, the letters a and b cannot be read.

Ans: As advised, in figure 7, the letters a and b are modified and is given below for your kind reference:

- The authors in the abstract declare that: increased level of 4-EPS in human body has also been found implicated in anxiety; in the conclusion they declare that: This technique will help in treatment of autism in patients. There is no linearity.

Ans: Thank you for your query; to prove the link between both statements, answer is given for your reference and is added to the modified abstract. This interaction study (4-EPS and BSA) may be helpful in devising strategies for the treatment of chronic kidney disease and other neuro related diseases, by producing synthetic compound that competes with albumin binding sites to allow 4-EPS clearance from the body. 

Reference:

1. Moradi M, Soleymani J, Tayebi-Khosroshahi H, et al (2021) Simple determination of p-cresol in plasma samples using fluorescence spectroscopy technique. Iran J Pharm Res IJPR 20:68

2. Li S, Tonelli M, Unsworth LD (2022) Indoxyl and p-cresol sulfate binding with human serum albumin. Colloids Surfaces A Physicochem Eng Asp 635:128042

3. Dezhampanah H, Mohammadi A, Mousazadeh Moghaddam Pour A (2021) Investigation on intermolecular interaction of synthesized azo dyes with bovine serum albumin. J Biomol Struct Dyn 1–12

4. Dezhampanah H, Firouzi R (2017) An Investigation on intermolecular interaction between Bis (indolyl) methane and HSA and BSA using multi technique methods. J Biomol Struct Dyn 35:3615–3626

5. Sajwan RK, Solanki PR (2022) A hybrid optical strategy based on graphene quantum dots and gold nanoparticles for selective determination of gentamicin in the milk and egg samples. Food Chem 370:131312

Reviewer #2: In the manuscript “Interaction of 4-ethyl phenyl sulfate with BSA: Experimental and molecular docking studies” the authors study the interaction of the e4-ethyl phenyl sulfate with the bovine serum albumin.

In my opinion, this work should be of interest to the readers of the Journal but I recommend the publication of this work after a major revision. In particular:

1) The title of the article contains the abbreviation BSA and not its full name. Please, the authors modify it.

Ans: As advised, suggested changes are made in the title and given below for your ready reference: Interaction of 4-ethyl phenyl sulfate with bovine serum albumin: Experimental and molecular docking studies.

2) In the abstract, is not clear the main aim of the article.

Ans: As suggested changes have been made in the abstract and is given below for your ready reference:

4-ethyl phenyl sulfate (EPS), a protein-bound uremic toxin found in serum of patients suffering from autism spectrum disorders (ASD) and chronic kidney disease (CKD). As per recent advances in the field, gut metabolites after their formation goes to blood stream crosses blood brain barrier and causes neuro related problems. Increased levels of 4-EPS in human body causes anxiety in patients and its role remains elusive. 4-EPS interacts with serum albumin in human body and thus, a model study of interaction of BSA with 4-EPS is presented in support of it. Absorption spectroscopy result demonstrated decrease in bovine serum albumin (BSA) absorption upon interaction with increasing concentration of EPS in a range from 2 μM to 100 μM. Moreover, this interaction was confirmed by the fluorescence quenching in presence of metabolite. The change in secondary structure was demonstrated by circular dichroism, synchronous fluorescence and Fourier transform infra-red spectroscopy. Docking studies reveals binding score of −5.28 Kcal mol−1, demarking that 4-EPS is involved in interaction with BSA via amino acid residues, forming the stable complex. This interaction study may be helpful in devising strategies for the treatment of chronic kidney disease and other neuro related diseases, by producing synthetic chemical that competes with albumin binding sites to allow 4-EPS clearance from the body. 

3) The introduction section is too long and it is difficult to easily understand the aim of the work. The authors declare that the selected compound is associated with autism disorder so, why the authors study this interaction with BSA and not with HSA? Please, the authors clarify it.

Ans: As advised, the introduction is modified and made crisp, to majorly focus on the proposed work. The modified introduction is incorporated in the revised manuscript.

We conducted this study with BSA due to its ready availability in laboratory and structure similarity with HSA, a bulky proteinaceous molecule. 

4) The sentence in paragraph 2.2 “confirmation of the successful…studies” should be removed because is not needed in the MM section.

Ans: As advised, the statement is removed from the MM section.

5) In the MM the authors include paragraph 2.2 defined “conjugate preparation” in which they described the sample preparation (BSA + 4-ethyl phenyl sulfate) before the spectroscopy characterization. The authors describe the incubation of the BSA with different analyte concentrations. The protocol reported is not clear. Please, the authors clarify it.

Ans: As advised, the language is refined to bring clarity in the work and modified line given here for your reference: the complex of 4-EPS with BSA was formed in molar ratios by keeping fixed concentration of BSA (2 μM) and 4-EPS concentration was varied to 2 μM (1:1), 10 μM (1:5), 20 μM (1:10), 50 μM (1:25), 100 μM (1:50), 150 μM (1:75), 200 μM (1:100).

The statement is incorporated in the revised manuscript.

Why the authors perform the incubation with an increased concentration of the analytes at pH 4.3? Have the authors studied the effect of this pH on the BSA structure? What is the effect on the binding? Have the authors information on the binding at neutral pH instead of low pH? Please, the authors clarify it.

Ans: According to one of the references, Milad Moradi et. al. suggest that PCS fluorescence intensity is enhanced in acidic pH [1]. BSA structure showed no degradation in its structure in low pH

---

## [Editor Report · Decision Letter 1]

6 Aug 2024

Interaction of 4-ethyl phenyl sulfate with bovine serum albumin: Experimental and molecular docking studies

PONE-D-24-20499R1

Dear Dr. Payal Gulati,

We’re pleased to inform you that your manuscript has been judged scientifically suitable for publication and will be formally accepted for publication once it meets all outstanding technical requirements.

Kind regards,

Sabato D'Auria

Academic Editor

PLOS ONE
---

## [Editor Report · Acceptance letter]

29 Aug 2024

PONE-D-24-20499R1 

PLOS ONE

Dear Dr. Gulati, 

I'm pleased to inform you that your manuscript has been deemed suitable for publication in PLOS ONE. Congratulations! Your manuscript is now being handed over to our production team.

Kind regards, 

on behalf of

Dr. Sabato D'Auria 

Academic Editor

PLOS ONE